# 'Our culture prohibits some things': qualitative inquiry into how sociocultural context influences the scale-up of community-based injectable contraceptives in Nigeria

Oluwaseun Oladapo Akinyemi [1,2] Bronwyn Harris,[3,4] Mary Kawonga[5]

For numbered affiliations see end of article.

**Correspondence to**
Dr Oluwaseun Oladapo Akinyemi;
ooakinyemi@comui.edu.ng

## ABSTRACT

**Objectives** To explore how sociocultural factors may support or impede the adoption of community-based distribution of injectable contraceptives in Nigeria.

**Design** A qualitative study based on inductive thematic analysis was conducted through in-depth interviews and focus group discussions.

**Setting** Most participants lived in Gombe State, North-East Nigeria. Other participants were from Ibadan (South-West) and Abuja (Federal Capital Territory).

**Participants** Through seven key informant interviews, 15 in-depth interviews and 10 focus group discussions, 102 participants were involved in the study.

**Methods** This study conducted in 2016 was part of a larger study on scale-up of community-based distribution of injectable contraceptives. Qualitative data were collected from traditional and religious leaders, health workers and community members. The data were audio recorded, transcribed and analysed using a thematic framework method.

**Results** Sociocultural challenges to scale-up included patriarchy and men's fear of losing control over their spouses, traditional and religious beliefs about fertility, and myths about contraceptives and family planning. As a result of deep-rooted beliefs that children are 'divine blessings' and that procreation should not be regulated, participants described a subtle resistance to uptake of injectable contraceptives. Since Gombe is largely a patriarchal society, male involvement emerged as important to the success of meaningful innovation uptake. Community leaders largely described their participation in the scale-up process as active, although they also identified the scope for further involvement and recognition.

**Conclusion** Scale-up is more than setting up health sector implementing structures, training health workers and getting innovation supplies, but also requires preparedness which includes paying attention to complex contextual issues. Policy implementers should also see scale-up as a learning process and be willing to move at the speed of the community.

## INTRODUCTION

Scale-up of effective health innovations, through their increased availability to a larger population over a wider geographic area,

### Strengths and limitations of this study

► The study participants represented a range of stakeholders — users of injectable contraceptives, community members, providers and health system managers.

► The study provided stakeholders who participated a distinct experience and some observations to reflect upon since the study was based on a current programme, rather than speaking merely about theoretical responses they or their community members may have to a health intervention.

► Our results highlighted that scale-up is influenced by several sociocultural factors; thus, showing the importance of paying attention to complex contextual issues during innovation uptake.

► The findings of our study emphasised how health systems and communities should interact in order to ensure successful scale-up of health innovations.

► Social desirability bias is a possibility as some of the participants in this study were involved in the community-based distribution intervention.

is essential for achieving universal health coverage, particularly in low and middle-income countries (LMIC).[1 2] Scale-up is influenced by various factors related to the innovation, the users and the social context in which the innovation is introduced.[3] The nature of the innovation itself, preferences and characteristics of the intended users (including their willingness to adopt the innovation) and ways that the innovation is adapted to the local context are also some of the factors affecting innovation scale-up.[4]

The sociocultural context, which includes religious and cultural beliefs, gender and societal norms and attitudes, is an important predictor of adoption of a health innovation in a population.[2 5] The sociocultural context may promote health or be a barrier to health-care utilisation.[6] For example, cultural beliefs

and practices have been shown to influence the uptake of a range of health innovations including male circumcision,[7] breast cancer screening[6 8] and contraceptives.[9 10] Without due consideration of the sociocultural context, health innovation development, delivery and scale-up are likely to be limited, and this may lead to waste of scarce health resources including time.[11] According to Bradley *et al*,[3] assessing the receptivity and willingness of community members to adopt a health innovation within their environmental and sociocultural settings is one of the key first steps in the scale-up process. In their conception, poor understanding by implementers of potential users' sociocultural context will limit scale-up efforts, while good understanding will help programme managers to recognise contextual factors that may enhance or inhibit the uptake of an innovation.[3]

Evidence highlights the influence of sociocultural context on the adoption of contraceptive innovations. For example, in a study in rural Poland, Colleran and Mace[12] reported that religious and sociocultural factors, particularly social networks, influence the uptake of contraceptive use more than individual characteristics. Dalaba and colleagues[13] in their study in northern Ghana reported that inadequate male participation led to resistance towards female reproductive freedom. Interventions at societal level, such as adequate male mobilisation coupled with appropriate community-based distribution (CBD) strategies, were shown to be effective in improving contraceptive uptake.[13] In Nigeria, where society is largely patriarchal, with consequences for access to and use of contraceptives, a study found that about two-thirds of rural women who had spousal approval were reported to use contraceptives while all those who were forbidden by their husbands did not use any family planning method.[14] According to the last Demographic and Health Survey in Nigeria, contraceptive prevalence rate (CPR) was 14.3% while only 2.4% of women of reproductive age group use injectable contraceptives. Understanding whether and how the sociocultural context influences scale-up of contraceptive innovation is thus important.

Evidence suggests that CBD of contraceptives is still needed in LMICs particularly rural or isolated communities in order to increase CPR.[15–17] The full potential of CBD to improve CPR has not been fully realised in sub-Saharan Africa.[18] Nigeria started its family planning policy review in 2007 and successfully piloted the CBD of injectable contraceptives through community health extension workers (CHEW) as a new innovation in the health system in 2010 in Gombe State, north-eastern region.[19 20] Afterwards, the scale-up of this innovation started in 2014, led by a national non-governmental organisation and supported by international donor agencies.[21]

However, the understanding of how sociocultural contextual factors may facilitate or impede the successful scale-up of the contraceptive innovation beyond the initial pilot site in Nigeria is limited. This study took place after scale-up started, thus it adopted both a retrospective slant to assess factors that affected scale-up and forward-looking approach to what may affect future scale-up effort (see study timeline in figure 1). The objective of this study is therefore to explore how sociocultural factors may influence the scale-up of the CBD of injectable contraceptives in Nigeria.

## METHODS
### Study design and setting
This cross-sectional study conducted from September to November 2016 was part of a larger research project[22 23] which explored the interplay between barriers and facilitators of scale-up of the CBD of injectable contraceptives guided by the AIDED model[3] (with components: Assess, Innovate, Develop, Engage, and Devolve) in order to inform wider scale-up of this innovation in Nigeria. The project took place in Gombe (north-eastern Nigeria), one of 36 states in Nigeria and a predominantly Muslim community. Gombe State, with a population of 2 353 879 people according to the 2006 population census,[24] has one of the lowest CPRs in the country (3.5% for modern methods and 4.0% for any method) and one of the highest maternal mortality rates (1726/100 000).[19 25 26] Gombe was purposively selected because it was the setting for the national pilot test for CBD of injectable contraceptives in 2010. Gombe is divided administratively into 11 local government areas (LGA). The study was conducted in two LGAs—Gombe (A) and Yamaltu/Deba (B). Although participants were primarily based in Gombe State, one interview took place in another state (Ibadan in the South-West) and three interviews in Abuja at national level (Federal Capital Territory).

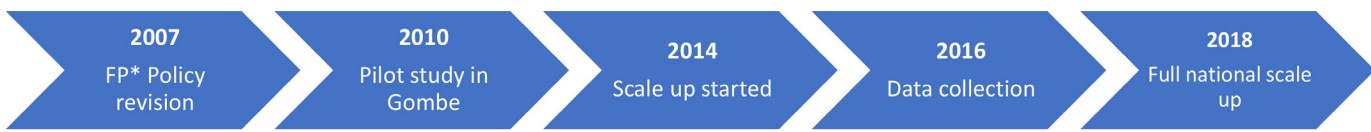

*Family planning

**Figure 1** Situating data collection within the timeline for pilot study and scale-up.

**Table 1** Participants' sociodemographic characteristics and data collection methods across study sites

| Location | Type | Interviews/ FGDs (n) | Participants (n) | Gender | | Participant type |
|---|---|---|---|---|---|---|
| | | | | M n (%) | F n (%) | |
| Federal and state level | | | | | | |
| | KII | 4 | 4 | 2 (50.0) | 2 (50.0) | NGO programme manager, senior official at the Federal MoH, NGO programme managers |
| Gombe State | | | | | | |
| LGA A | KII | 3 | 3 | 2 (66.7) | 1 (33.3) | Senior officials of State MoH. |
| | IDI | 7 | 7 | 3 (42.9) | 4 (57.1) | Health workers (nurses, CHEWs), traditional and religious leaders |
| | FGD | 5 | 40 | 16 (40.0) | 24 (60.0) | Current contraceptive users, older women non-users, younger women non-users (in reproductive age group), married and single men |
| LGA B | IDI | 8 | 8 | 4 (50.0) | 4 (50.0) | Health workers (nurses, CHEWs), traditional and religious leaders |
| | FGD | 5 | 40 | 16 (40.0) | 24 (60.0) | Current contraceptive users, older women non-users, younger women non-users (in reproductive age group), married and single men |
| Total | | 32 | 102 | 43 (42.2) | 59 (57.8) | |

CHEW, community health extension worker; FGD, focus group discussion; IDI, in-depth interview; KII, key informant interview; LGA, local government area; MoH, Ministry of Health; NGO, non-governmental organisation.

## Participants and sample

In total, 102 people participated in this study (see table 1). The participants were selected through a purposive sampling method, where study respondents were recruited based on characteristics of interest, availability and ability to provide relevant information[27 28] about the research question: how may sociocultural context influence the process of introducing, translating and integrating health innovation? Participants included a range of stakeholders, including traditional and religious leaders, health workers as well as community members. One focus group discussion (FGD) each was conducted with different types of participants who were involved and could give an account based on their experiences of the programme with the aim of exploring qualitatively the views, concerns from this range of key role players, not aiming to quantify, compare or rank frequency of views. Each of the FGDs had eight participants. The selected participants were therefore deemed appropriate role players to share their own experience and views as well as reflect on and summarise the views of others about the implementation of the CBD of injectable contraceptives within their sociocultural context. The number of interviews and FGDs was determined by the principles of saturation, a point where no new information is emerging,[29 30]

and availability of resources as well as study time.[31] The interviews and FGDs were conducted by the first author assisted by three trained research assistants.

This study adopted the inductive thematic analysis. This approach was guided by available literature as well as insights from the participants—codes were generated from key themes identified from transcripts.[29 32–34]

## Data collection

Data were collected through 7 key informant interviews (KII), 15 in-depth interviews (IDI) and 10 FGDs. KIIs, IDIs and FGDs were conducted with senior stakeholders, healthcare workers and community members, respectively. Guides were used to conduct the interviews and FGDs (see online supplementary file 1). All guides were translated to the predominant local language—Hausa. Interviews with key stakeholders covered issues of programme planning and design, community entry and resolution of challenges with implementation. However, FGDs dealt with issues of programme implementation, community members' preferences with innovation delivery and packaging. Nonetheless, some issues covered in the interviews and discussions (FGDs) were similar. Interviews and discussions were tape recorded and later transcribed in full and translated where necessary. Transcription was

done using a protocol which emphasises verbatim transcription of recordings. Both transcription and translation were done by the same research associate who is vast in both Hausa and English languages. Back translation was subsequently done by another associate vast in the two languages; the back translations were afterwards compared with the original transcripts in order to ensure that the original meaning is retained.[35]

## Data analysis

Transcripts were analysed iteratively with the aid of the NVivo (V.10) software using the inductive thematic approach.[29 30 33 34] Coding was done by the first and last authors independently at first before analysis started; these codes were then compared and harmonised. Emerging codes from the transcripts were added to predefined themes: community members' perceptions regarding CBD of injectable contraceptives, community members' acceptance (or rejection) of contraceptives and community engagement in the scale-up process. Emerging themes were indexed and compared with themes from subsequent interviews until saturation was attained. A codebook was developed after all the data were collected; this was then applied to the transcripts. Although numbers of interviews and FGDs were set at the outset based on empirical guidance, however during analysis, it was confirmed that saturation,[29] where no new information was obtained, was actually reached before the cap was reached in both interviews and FGDs. Analysis progressed alongside data collection. The Standards for Reporting Qualitative Research reporting guidelines for qualitative studies[36] were used in preparing this manuscript.

## Patient and public involvement

The study participants were involved in shaping the study particularly during the pretest which led to the modification of some ambiguous questions in the guides. Also, some participants helped with the recruitment of other respondents through a snowball sampling[37] procedure. Furthermore, findings from this study and the larger research project have been presented in conferences with useful feedback; a policy brief has been developed to be disseminated to policymakers.

## RESULTS

Of the 102 study participants, 59 (57.8%) were female (table 1) with a mean age of 42.8±14.2 years. Similarly, mean ages of FGD and interview participants were 42.7±14.6 and 46.7±6.0 years, respectively. Six of the 10 FGDs were made up of female participants, each FGD comprising eight respondents. Participants described a number of sociocultural factors that challenged the scale-up process right from the pilot stage; these include traditional and religious norms about fertility, perception about family planning including adverse effects as

well as patriarchy and men's fear of losing control of their spouses (figure 2).

## Deep-rooted cultural and religious beliefs about fertility

Interview participants spoke about a deep-rooted belief among community members that children are 'divine blessings' and that procreation should not be regulated. Related to this world view, participants described a subtle resistance to the uptake of injectable contraceptives, more passive than overt in its expression by individuals and families who would feign acceptance but refuse the intervention when given.

> …there will always be resistance to this use of contraceptives, because here, we are people that believe that children are a blessing, so you give birth to as many as possible. …The resistance may not be as a group, but when you come to individual levels… as a government, they might not show you the resistance, but when it comes to the actual usage, that's where you start having problems. (IDI, male, doctor, SMoH)

Also, there were issues about religious beliefs influencing perception about family planning/contraceptive use at the community level. Religiosity, that is, strong adherence to religious beliefs, was said to be one of the major factors impeding the uptake of contraceptives in general.

> …when you look at northern Nigeria, majority are Muslims and when you look at the south eastern states, you know we have issue of religion, issue of uptake of all contraceptives, not even injectables alone. To the south eastern man, most of them are Catholics, they frown at family planning, so also here (northern Nigeria) people frown at family planning in general. (KII, male, senior official, SMoH)

## Awareness and perceptions regarding family planning

Most interview respondents agreed that there was a need to continue to educate community members about the overall benefits of contraceptives and family planning as well as emphasise possible side effects of contraceptive use among women. Also, informants were of the opinion that lack of knowledge about benefits of contraception contributes to reluctance to adopt the innovation. In the words of a respondent:

> …the community and people at the grassroots have little information (on contraceptives). (KII, female, senior official, SMoH)

These challenges were further compounded by reports of adverse effects of the contraception from some women especially '*severe bleeding during menstruation*' (IDI, female, CHEW). Healthcare providers felt that to effectively deal with these challenges, it was important to listen sympathetically to the women and treat reported side effects, in line with the training received for the innovation scale-up. Education was also given when clients had questions

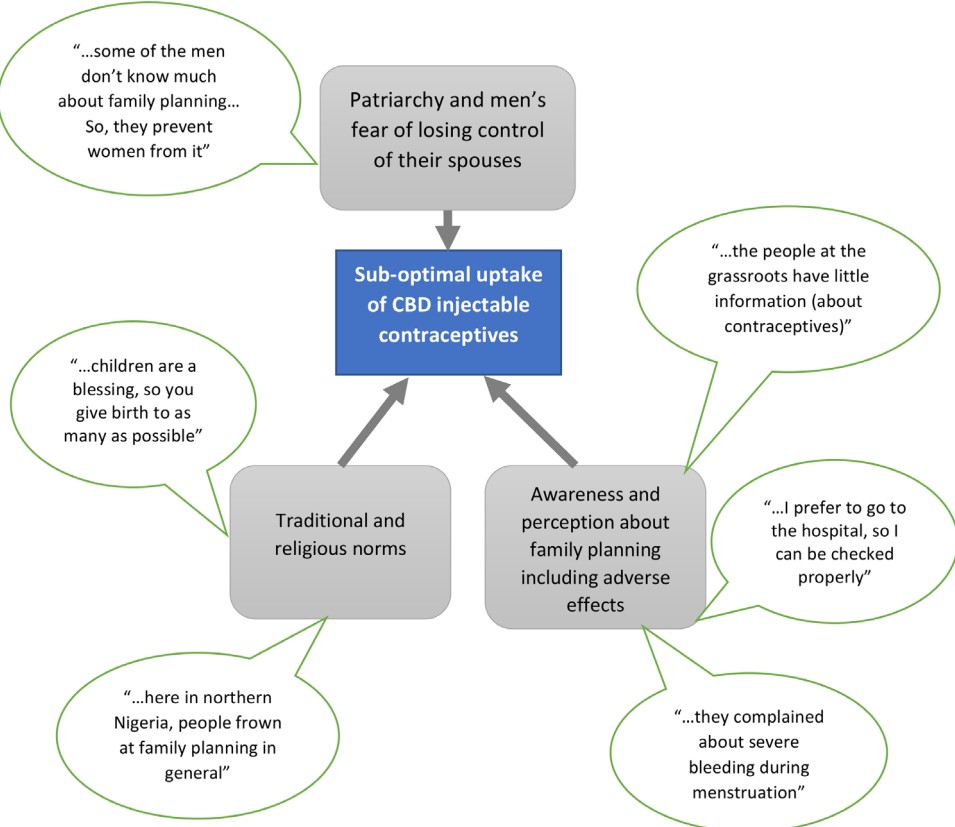

**Figure 2** Sociocultural factors influencing uptake of community-based distribution (CBD) of injectable contraceptives in Gombe State.

in order to resolve doubts about the innovation. For example, a respondent said:

> We listen to them, and we explain to them that since it (contraceptive) is a new thing in the body, they must experience such things… (IDI, female, nurse)

Focus group discussants highlighted many reasons why women in their reproductive age group used injectable contraceptives, including desire to stay attractive in order to prevent husbands from 'straying', space the children and enjoy life with fewer financial and parenting constraints. One of the discussants who currently uses injectable contraceptives described her motivation thus:

> …If we space our children well, we will enjoy our lives because our husbands will always desire us since we are always looking attractive. He will not stay out late because he will be thinking of you at home. (FGD, female, current user)

Also, a participant stated that the impetus for contraceptive use was the fact that spacing her children afforded her the ability to take care of herself and the home thereby promoting hygiene and healthy living in the home. She said:

> …You will be able to take care of yourself as well as your house, your husband will also desire you as you will look good as well as your children. Flies will not follow you around and anyone that comes to your

house will be able to eat and drink because everywhere is clean and attractive. (FGD, female, current user)

### Perceptions regarding the CBD approach

During the FGDs, women who were current users of injectable contraceptives stated their preference for the community-based approach over the facility-based system of distributing the contraceptives. Some of their reasons for this position included convenience, affordability (saves cost as transport fare is not needed), improved privacy and focused attention from the health worker.

> …During home visit, it's between you and the woman (health worker) and she can explain it very well to you since she is not in a hurry; although in the hospital they can also explain but the home visit is better since the health worker will explain until you understand. (FGD, female, current user)

Likewise, some male discussants preferred the community-based approach because they believed it was more confidential and gave women more time to deal with domestic issues. The CBD approach was also believed to align with men's desire to be in control of 'family issues'.

> …Maybe the time the wife wants to go (to the health facility) the husband needed her for something else… Sometimes some husbands want all these things to be

confidential, but when you now tell them to release their wives to the hospital, they will think you are exposing the family issues. (FGD, married men)

However, one of the discussants who favoured the facility-based approach '*prefer[ed] to go to the hospital, so I can be checked properly, including to check my BP*' (FGD, female, current user). Similarly, some married men also preferred the facility-based approach as they believed that the community-based method is intrusive of their privacy and undermines their authority:

…My advice is that the woman should come to hospital …because some husbands will query you that who gave you the permission to see their wives (at home) and advise them about child spacing. Some disagree with the community-based method. (FGD, married men)

### Spousal involvement in family planning
In addition, male involvement was highlighted as important if uptake of innovation will be meaningful since Gombe is largely a society where the women need the men's permission for doing many things including obtaining injectable contraceptives.

Some men will not accept it (contraceptive) because they want plenty children…she (the wife) should continue begging the husband until he agrees if she wants to use contraceptives. (FGD, women not using contraceptives)

Furthermore, cultural beliefs and sociocultural context were reported as fuelling poor awareness and knowledge about contraceptive use. Also, male involvement was described as important in dealing with the problem of ignorance and poor perception about family planning in the community. According to the participants in this study, once the men are well informed, it is easier to get their buy-in and cooperation in permitting their wives to get the injections.

…some of the men don't know much about family planning commodities and some (people) are telling them some of the side effects. So, they prevent the women from using them…that's why we need more sensitization to the men and the youths. (IDI, female, nurse)

The perception of men (about contraceptives) is lower than that of women. The women know the implications better than the men. The men in the rural areas don't mind having any number of children… (FGD, married men)

Also, a female health worker emphasised the importance of male involvement particularly in communities where the literacy rate is low:

Honestly, we need to do more in carrying them [men] along. Male involvement is very important because you know they are our superiors. We in the North,

it is only the educated that may be sometimes at loggerhead (with their husbands) … but these illiterates and native people, it is always the man that is superior. So, it is very important for us to consider male involvement, it will help a lot. (IDI, female, nurse)

### Community support for scale-up
The issue of community support for scale-up was solely explored with community and religious leaders. The traditional leaders interviewed believed that community support for the innovation is very high.

In fact, the support our community gives to this programme, without exaggeration, I will say maybe 80%–90%. They (the community) have been very supportive of the programme. (IDI, male, traditional ruler, Yamaltu/Deba)

Definitely, you have 70% support from the communities. From the religious leaders, traditional rulers and even of ward heads in the community we get 100% support. (IDI, male, traditional leader, Gombe)

However, some cautioned that the community support is not as high as expected due to what they call low knowledge and awareness about contraceptive choices in Northern Nigeria.

…Really, people have supported the program but I know some people do not have the knowledge (about injectable contraceptives) but those that have the knowledge are really in support of the program… [you know] the Northern culture prohibits some things, that's why I said the support for it (injectable contraceptives scale up) is not as much as we expect. (IDI, male, Christian religious leader)

One community leader described his role as active in the scale-up process because he helps in disseminating information to all the wards under his domain. He shared his experience:

Whenever the need arises, we normally call the kingmakers, district head kingmakers, they are about seven, and we have at least 54 ward heads, we use to call them from time to time and enlighten them on the programme. In fact, almost all the community leaders are fully aware of the programme. (IDI, male, traditional leader)

Many of the leaders ascribed the progress with the scale-up in their communities to the involvement of the traditional leaders in the programme. However, some leaders complained of not being officially involved in the programme, although they still offer their support.

There are so many traditional rulers that were involved in this program; that is the reason why it has been accepted 100%, especially in Gombe State now. (IDI, male, traditional leader, Gombe)

…My involvement is…passive involvement. Of course,

we preachers, you can push people, encourage them in your own little way... We preach it, we tell people about it... (IDI, male, Christian religious leader)

## DISCUSSION

In our study, there were a number of sociocultural issues alluded to as factors that may affect scale-up of a contraceptive innovation in Nigeria, including patriarchy, traditional and religious norms, beliefs and myths about contraceptives and family planning. Agunbiade and Ogunleye[38] reported in their study assessing the sociocultural barriers to the scale-up of exclusive breast feeding in Nigeria, the scale-up process cannot be separated from the context and the culture in the environment. Further, in a study among African–American women, culture and health beliefs were found to be significant predictors of health innovation uptake.[39]

Our study indicates it is vital to secure the understanding and support of the men and significant others for scale-up of the CBD of injectable contraceptives to succeed in environments like our study site that are still largely patriarchal.[38] Similar to the findings of our study, male involvement and support of women's significant others, particularly the mothers-in-law, have been reported to be crucial to innovation scale-up in Nigeria as these significant others have strong influence on choices made by women.[38] In other settings, increased male participation through increasing men's knowledge and reducing their opposition has been shown to improve uptake and continued use of contraceptives[40–42] and enhancement of prevention of mother to child transmission of HIV scale-up processes.[43] In this study, one of the reasons why women reportedly use contraceptives was to look attractive and satisfy their husbands in order to prevent them from engaging in extramarital relationships. Thus, contraception is both liberating for women on one hand but also perhaps serving the very patriarchy they are trapped in on the other. Studies have shown that in many African societies, patriarchy is perpetuated by the socialisation process which starts from the family and is often preserved by women themselves.[44 45]

Also, findings from our study show passive resistance to contraceptive uptake, particularly at the individual levels, by both women and their spouses. Research has shown that passive resistance is one of the major reasons for the high failure rate of innovation scale-up.[46 47] Passive resistance often results from individuals' resistance to change and contentment with or lack of power to change the status quo.[47] Further, consumers will not adopt an innovation if they are not sure of its value or benefits.[48] According to Heidenreich and Kraemer, passive innovation resistance should be addressed before the introduction of new innovations into the society.[46] Strategies for reducing passive resistance to uptake of health innovations in the community include actively involving men in their uptake,[13] as well as giving adequate and correct information about the advantages of contraceptive use

to all stakeholders—health workers, community and religious leaders, trained volunteers and programme managers.[48]

Since the passive resistance reported in this study is largely underpinned by deep-seated issues of patriarchy, religious beliefs and traditional norms/values, it is imperative for policymakers and programme managers to engage men—who are custodians, in most cases, of both religious and cultural values.[49 50] Male involvement can bring benefits for all community stakeholders,[49] as well as social change.[51] With careful management and ongoing monitoring, the involvement of men may ultimately give the scale-up of this health innovation a greater chance of success.

Furthermore, religious beliefs were also found to influence the scale-up of the community-based injectable contraceptives and family planning in general in our study sites in Northern Nigeria. Religion has been shown to significantly influence decisions on contraceptive use,[52] as well as individual/household choices, lifestyle and even traditional values.[53] Thus, religion has great implications for the social marketing of commodities particularly contraceptives, as uptake of health products has been shown to differ significantly among adherents of major religions.[53] For example, research done in Islamic countries[54 55] highlights the need to understand Islamic religion in order to do effective social marketing, and recommends that approval be obtained from religious leadership before certain sensitive products like contraceptives are marketed. Understanding religious nuances in local contexts is important in multi-faith countries such as Nigeria—for example, what 'works' in the largely Islamic North[56] may need adaptation for Christian populations elsewhere.

Our study demonstrated the importance of strong community support and partnership with community gatekeepers in the scale-up process. Research has shown that the CBD approach cannot succeed without community participation.[57] Therefore, programme implementers need to learn from the community what works (or may work) in the setting, what distribution method the community prefers or what modification to the existing methods they favour. Research has shown that there is no quick fix to scaling up.[57] It is essentially a learning process which is unique in different contexts. Therefore, scale-up may not be effective without adequate contextual knowledge and willingness to learn through the process of community participation and stakeholder collaboration.[57 58]

While the focus of this paper has been on challenges as identified in the wider context of religion and patriarchy, many of the suggested solutions rely on the health systems and those working in it, to navigate these challenges and overcome them. However, this cannot be a static or reactive approach but rather needs to recognise that health systems and those working in them are also part of the sociocultural context and therefore it is important to find ways to support healthcare workers in mediating, translating and promoting the innovation in this context.

## Limitations of the study

Some limitations of this study should be emphasised. As with any qualitative study, the findings are not statistically generalisable but the theories and ideas can be applied to other contexts. Also, some of the participants in this study were involved in the CBD intervention, therefore they may provide more socially desirable answers. However, a strength of this study is that it provided stakeholders who participated a distinct experience and some observations to reflect on since the study was based on a current programme, rather than speaking merely about theoretical responses they or their community members may have to a health intervention. Thus, the findings of this research provide valuable information into how sociocultural context and user support affect the scale-up of health innovations in LMICs, particularly in a patriarchal society like Nigeria.

## CONCLUSION

Our study suggests the need for continuous community education and engagement with the significant others and community leadership structures—both traditional and religious. Thus, providing an innovation does not necessarily translate to uptake. Therefore, programme implementers should also consider sociocultural and health system contexts when packaging a health innovation. Hence, supporting those in the health system to mediate the innovation in ways that are sensitive to the context but which, by working with communities and others, promotes transformation is imperative. In addition, incorporating the CBD approach with other outreach programmes, developing a more effective supervision of the approach may help make CBD more productive. Interventions to improve male participation will likewise go a long way to improve the scale-up process. Policy implementers should also see scale-up as a learning process and be willing to move at the speed of the community.

**Author affiliations**

[1]Health Policy and Management, University of Ibadan College of Medicine, Ibadan, Oyo State, Nigeria
[2]Community Health, University of the Witwatersrand School of Public Health, Johannesburg, South Africa
[3]Division of Health Sciences, Warwick Medical School, University of Warwick, Coventry, UK
[4]Centre for Health Policy, School of Public Health, University of the Witwatersrand, Johannesburg, South Africa
[5]School of Public Health, University of the Witwatersrand, Johannesburg-Braamfontein, Gauteng, South Africa

**Acknowledgements**  We are grateful to Drs Mariya Saleh, Wole Adefalu, Edward Oladele and Hadiza Kamofu for expediting access to key informants. Also, we appreciate Dr Alex Eze for his feedback on the early draft of the manuscript. We are grateful for the support of Dr Busola Adebayo, Jimi Latunji and Wunmi Folajimi-Senjobi with editing the manuscript. We thank all the participants in this study.

**Contributors**  OOA conceived the study. OOA and MK designed the study. The data were collected by OOA. OOA, MK and BH drafted and commented on the draft of the manuscript. OOA, MK and BH read and approved the final version of the manuscript.

**Funding**  This research was supported by the Consortium for Advanced Research Training in Africa (CARTA). CARTA is jointly led by the African Population and Health Research Center and the University of the Witwatersrand and funded by the Carnegie Corporation of New York (Grant No: B 8606.R02), Sida (Grant No: 54100113), the DELTAS Africa Initiative (Grant No: 107768/Z/15/Z) and Deutscher Akademischer Austauschdienst (DAAD). The DELTAS Africa Initiative is an independent funding scheme of the African Academy of Sciences' (AAS) Alliance for Accelerating Excellence in Science in Africa (AESA) and supported by the New Partnership for Africa's Development Planning and Coordinating Agency (NEPAD Agency) with funding from the Wellcome Trust (UK) and the UK government.

**Disclaimer**  The statements made and views expressed are solely the responsibility of the authors.

**Competing interests**  None declared.

**Patient and public involvement**  Patients and/or the public were not involved in the design, or conduct, or reporting, or dissemination plans of this research.

**Patient consent for publication**  Not required.

**Ethics approval**  Ethical approval for this study was obtained from the University of Ibadan/University College Hospital Ethical Review Board (Reference No: UI/EC/16/0022) as well as the Human Research Ethics Committee (Medical) of the University of the Witwatersrand (Reference No: M160737). Written informed consent were obtained from study participants before the interviews and FGDs (see Supplementary file 2). Participants were also assured of confidentiality by ensuring that all identifiers were removed from the data and that only the research team has access to the data.

**Provenance and peer review**  Not commissioned; externally peer reviewed.

**Data availability statement**  Data are available in a public, open access repository. Extra data can be accessed via the Dryad data repository at http://datadryad.org/ with the doi: 10.5061/dryad.6q573n5w8.

**ORCID iD**
Oluwaseun Oladapo Akinyemi http://orcid.org/0000-0003-4135-1459

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
