## [Reviewer comments · BMJ Open]

ARTICLE DETAILS

TITLE (PROVISIONAL)	“Our culture prohibits some things”: Qualitative inquiry into how sociocultural context influences the scale up of community-based injectable contraceptives in Nigeria
AUTHORS	Akinyemi, Oluwaseun; Harris, Bronwyn; Kawonga, Mary

VERSION 1 – REVIEW

REVIEWER	Djibril Ba Penn State College of Medicine, Hershey, PA 17033 USA
REVIEW RETURNED	19-Dec-2019

GENERAL COMMENTS	Comment: Thank you for submitting this manuscript. The manuscript by Akinyemi et al. provides results from an analysis of 15 in-depth interviews and 10 focus group discussions from 102 participants. The authors found that sociocultural challenges to scale up included patriarchy and men’s fear of losing control over their spouses, traditional and religious beliefs about fertility, and myths about contraceptives and family planning. While in overall I think the manuscript is well written. However, I do not think the manuscript is scientifically strong enough to be published in its current stage. The authors didn’t report any descriptive statistics or effect size. It hard to understand what the study is about and the objectives. The authors didn’t also reported the overall rate of contraceptive use in Nigeria and the rate of injectable contraceptive. Line18-19: Of the three authors.....this sentence is unnecessary.
---

REVIEWER	Emily Namey FHI 360, USA
REVIEW RETURNED	30-Dec-2019

GENERAL COMMENTS	This manuscript identifies a number of socio-cultural factors that have affected scale up of community-based delivery of injectable contraceptives in one region of Nigeria. The findings are consistent with those of research on uptake of innovation in similar areas or adjacent health fields. A strength of the research is that it is based on an existing program, providing the stakeholders who participated a discrete experience and set of observations to reflect upon, rather than speaking solely about hypothetical responses they or their communities may have to an intervention. This strength could be more clearly explicated and highlighted in the manuscript.
--

	The primary weakness of the current draft of the manuscript is in the description of the methods and analyses, which are opaque and reference citations that do not show convincing appreciation for the empirical evidence or processes involved. I have provided specific questions and suggestions in the attached document. My primary concerns are around sample size - how was it decided upon and how can you bolster its justification? - and analysis approach - was this really grounded theory? Qualitative research has struggled for many years with a reputation for sloppy or "black box" methods which then relegates findings from qualitative research to "anecdotal" status for many scientific audiences. To avoid perpetuating this perception, I have recommended a major revision to the Methods and Findings sections of this manuscript, to improve the clarify of the former and confidence in the latter. BMJ Open review notes for "Our Culture Prohibits Some Things..." PAGE 4 Ln 22-24 "The sociocultural context may promote health or be a barrier to health care utilization." Ln 27-31 "Without due consideration of the sociocultural context, health innovation development, delivery and scale up is likely to be limited, if not dangerous!12" The claim of potential danger seems undeveloped; unclear whether the citation supports this. Suggest explaining briefly (dangerous to whom? Why?), re-wording, and/or removing the exclamation point. Perhaps move text from Page 5, lines 9-18, as these further explain the point/purpose. PAGE 5 2.1 Study design Was anything published on the larger study of which this research was a part that you could reference? In either case, consider adding another sentence about what that entailed, which might help to justify the sample size and selection. I'm curious about the timing of the study relative to scale up. Seems to have happened after scale up ... were there issues/problems that led to the study? Or just the way the funding worked out? Lines 22-25 seem to suggest that this is a forward-looking approach to what may affect scale up, but the timing suggests perhaps a retrospective look at what has affected scale up. Participants and Sample I'd like to see a clearer description in the narrative of which participants were interviewed and which in focus groups. The table provides detail which seems to indicate HCWs had IDIs, community members had FGDs, and senior stakeholders had KIs. Also, the table seems to suggest one FGD each with different types of participants. If that is the case, it may be a very lean sample from which to draw conclusions and more will need to be said about why this was the selected sampling stratification (e.g., you were aiming to quickly assess the range of potential concerns, rather than understand the relative frequency/saliency of issues). It seems many of the questions on the guides are asking participants to summarize the views of others, rather than offer
--	---

their own view or experience. This might be used as a rationale for conducting focus groups, which are typically helpful at getting at social norms.

Also, in describing the setting, since religious values are discussed extensively later, it would be helpful to know if this is a predominantly Muslim community as readers may assume.

PAGE 6

As above, I'd like to see a more robust discussion of sampling. Justification of sample size could be better cited; there are several empirical studies that help to establish adequate sample size to reach saturation that would strengthen the authors' case for whether/why the selected sample was sufficient. Guest et al. (2006 and 2016) provide tables summarizing empirical literature on sample sizes and present empirical data of their own for interviews and focus group sample sizes.

It would be helpful to the reader to call out the average (or minimum) size of focus groups.

It's possible (and common) to conduct an inductive thematic analysis without using grounded theory. Grounded theory is specifically indicated when the purpose of the research question is to develop a theory about a phenomenon. To my sense, the research question here is an applied, rather than theoretical, question: what are the specific issues or behaviors that could affect scale up? In this case, grounded theory is likely not warranted and my guess is that the constant-comparison method called for by grounded theory was not used. In any case, there are neither appropriate citations for grounded theory nor inductive thematic analysis. Check Glaser and Strauss, Charmaz, or Maman for grounded theory; check Guest et al. or Bernard for inductive thematic analysis.

2.2. Data collection

This paragraph isn't very clear in distinguishing what was asked of whom and in which format. Were community members involved in FGDs? Line 33 refers to "interview guides used for discussions with community members" which I imagine are focus groups and therefore, according to the sentence prior, followed focus group guides. You could just say "all guides were translated..." to avoid confusion. It would be helpful to clarify what was asked of all participants and what was asked specifically of one group or another. As it is, it seems all of the same topics were covered by all types of participants, which seems unlikely [supplemental file is helpful, but reader should have a grasp without reviewing those].

Ln 44-45 Was there any kind of transcription protocol? Were the translations and transcriptions done by the data collectors or different people? Helpful to know a little more about the process so the reader can get a sense of data accuracy and consistency across transcripts.

2.3 Data Analysis

I have briefly reviewed references 24 and 25 and neither seems to align with the description of the analysis process presented here. Specifically, article 24 relates to translation and reflexivity, and is very much a conceptual rather than processual piece. Article 25

outlines some of the tenets of grounded theory and walks through their use with CAQDAS, but it doesn't mention a "thematic framework approach". The description of pre-defined "thematic areas" sounds like they are not really thematic areas at all, but rather they reference the domains of inquiry, or what I would call "structural codes" – areas marked in the transcripts according to what was asked of the participant at that point.

This description of the analysis of emergent themes is somewhat weak. Who performed coding? At what level(s)? Was there any inter-coder reliability assessment?

The statement that "Interviews were stopped at the point of saturation, where no new information was obtained.²⁴" is unclear. Was the sample set at the outset, based on empirical guidance on sample size relative to saturation, or was the aim all along to sample until saturation was reached. Was analysis really progressing alongside data collection, or were all data collected and then analysis began? It is not clear from either the data collection or analysis sections what the sequence of events was.

Also, related to saturation, was it that no new information was gained from anyone? Was there any comparative analysis of the different groups? Was the reaching of saturation predicated on the FGDs alone?

PAGE 7

Lines 6-9 "The authors reviewed the transcripts and compared themes, before these were applied to subsequent coding of all the transcripts." This part of the process is also a bit unclear. Is this meant to say that transcripts were reviewed prior to the development of a codebook? Was a codebook then developed? After how many interviews/FGDs? Or was all data collected (same Q as above) and then a codebook developed? It seems unusual to declare saturation reached (and to stop data collection) before coding is completed.

RESULTS

I would like to see some summary of the demographic characteristics of the participants leading this section, particularly the FGD participants, so the reader has some way to interpret better from whom the subsequent findings/quotes come. For the KIIs and IDIs, it may suffice to reference broad general categories as in Table 1. For FGDs, gender and median age of participants would be useful.

Figure 1 is, I think, a summary of the three main overall findings. Each of the main areas to be discussed in the Results section should be identified in the narrative introduction to the section, so they align with the headings that follow and don't require the reader to process Figure 1 to understand what is coming.

3.1 – here and throughout, where you say "participants" indicate which type of participants. Was this shared among all categories? The quotes for the first several findings seem drawn only from IDIs/KIIs – was this because only they were asked or because these themes did not come up in FGDs?

Also throughout, do you have the possibility of indicating the frequency of statements in support of the finding you're

	presenting? In the cases where you indicate “one person said” that’s helpful to know; where it’s not specified, it’s hard to know if this was a frequently repeated theme or a one-off idea. Ln 52-54 copy edit “Emerging from this world view, participants described a subtle resistance” in this phrasing, the participants themselves were emerging from this world view, when I think you mean the resistance was related to the world view. For section 3.3. was this asked only of FGD participants? Could you do a comparative analysis of responses? Were current users generally in favor and men mixed? An issue is that the FGD is the unit of analysis, so you only have 10 and it’s hard to make determinations about whether a group is for/against something. In some sections, like 3.5, it seems only one category of participant is represented. Were only traditional/religious leaders asked these questions about community support? If so, state that clearly. PAGE 14 4.1 Limitations – This section should be revisited. The sample for this study was extremely small and, as yet, unjustified in its size and scope. “... findings are not statistically generalizable but the theories and ideas can be applied to other contexts.” It’s not clear that these theories and ideas can be applied in other contexts with any confidence, based on the study design and data alone... perhaps more accurately, the findings are in line with what others have found in similar (religious/LMIC) settings, which adds to the collective body of knowledge on socio-cultural issues that affect contraceptive uptake. 4.2 Recommendations “Further studies are needed on the barriers and facilitators to community-based distribution of health innovations.⁴¹ Although the pilot project¹⁶ provided some evidence that community based distribution is feasible in Nigeria, more understanding about what may impede or enhance this approach to health commodity distribution is needed.” This statement is fairly generic. It seems you cited a number of studies on barriers and facilitators, and listed several things that stakeholders need to consider. So what, specifically, are questions that need further research? PAGE 15 4.3 Conclusions There is a lot of repetition between the Discussion and Conclusion sections. Suggest focusing the Discussion section on situating the findings within the broader literature and the Conclusions on what the findings mean in terms of next steps for research and/or practice.
--	--

REVIEWER	Margaret Gichane RTI International, USA
REVIEW RETURNED	02-Jan-2020

GENERAL COMMENTS	This manuscript examined the sociocultural factors that influence the scale-up of community-based distribution (CBD) of injectable contraceptives. The main strengths of the article are that the
---

	authors had a large qualitative sample that represented important stakeholders (e.g. traditional and religious leaders, health workers and community members) and they utilized both individual and focus groups to gain a variety of perspectives. There are several areas that need to be strengthened. The introduction does not draw on any of the research that has been conducted examining CBD in other countries in Sub-Saharan Africa. The study is described as using a grounded theory approach, however the analysis described does not align with grounded theory. Finally, there are a few gaps in the limitations and recommendations. I propose the following changes: Abstract 1) Conclusion needs to more align with the findings. Include a sentence summarizing findings. Introduction 2) The introduction does not include any description on the current evidence on community based distribution of injectable contraceptives in other countries in Africa. Consider including: a. Prata, N., Gessesew, A., Cartwright, A., & Fraser, A. (2011). Provision of injectable contraceptives in Ethiopia through community-based reproductive health agents. Bulletin of the World Health Organization, 89(8), 556–564. https://doi.org/10.2471/BLT.11.086710. b. Prata, N., Vahidnia, F., Potts, M., & Dries-Daffner, I. (2005). Revisiting community-based distribution programs: are they still needed? Contraception, 72(6), 402–407. https://doi.org/10.1016/j.contraception.2005.06.059. c. Hoke, T., Brunie, A., Krueger, K., Dreisbach, C., Akol, A., Rabenja, N. L., et al. (2012). Community-based distribution of injectable contraceptives: introduction strategies in four sub-Saharan African countries. International Perspectives on Sexual and Reproductive Health, 38(4), 214–219. https://doi.org/10.1363/3821412. d. Gichane, M. W., Mutesa, M., & Chowa, G. (2019). Translating Evidence into Policy Change: Advocacy for Community-Based Distribution of Injectable Contraceptives in Zambia. Global Social Welfare, 6(1), 41-47. Methods 3) Include information on who conducted the interviews 4) Page 5, Line 32: Describe the parent project in more detail. Additionally, include which stage in the research study the interviews were collected. If it was after study activities had commenced include in the limitations how that may have affected findings. 5) Page 6, line 12: change “a point where no new data is being gotten” to “no new information is emerging.” 6) Page 6, line 30: Change, “Interview guide and focus group guide were used “ to “Guides were used...” 7) Page 6, line 52: The data analysis is described as a “thematic framework approach” but the study was classified as using a “grounded theory approach.” Grounded theory follows a specific set of steps when coding that were not part of this study. See research from qualitative methodologists such Charmaz, Strauss and Glaser. It is more appropriate to use just name this as a qualitative study that used a “thematic framework approach” for analysis. 8) Table 1. Though it is correct to say the “n” for FGDs refers to number of groups not individual participants it makes it difficult to compare the table to the text (n=32 interviews vs. 102
--	---

	participants). Consider including a column where you include ns for participants. For IDIs the n will be the same in both columns but for FGDs one column will have the number of groups and the other will have the total number of participants in groups. Results Well written. No additional comments. Discussion 9) Page 13, Line 15: Indicate who is passively resisting. Is it women? Husbands? 10) Page 14, line 27: It is unclear what finding this paragraph is related to. Which solutions were proposed? 11) Page 14, line 42: An additional limitation is that you recruited some participants who were involved in a community based distribution intervention therefore they may provide more socially desirable answers. 12) Page 14, line 54: These recommendations seem very general and not related to the findings. The results point to the importance of considering patriarchal views and religiosity when designing CBD programs. It seems like interventions with men should be recommended.
--	--

VERSION 1 – AUTHOR RESPONSE

Reviewer 1 (Djibril Ba)

1. The authors didn't report any descriptive statistics or effect size.

Response: Descriptive statistics and effective size are not hallmarks of qualitative study

2. It hard to understand what the study is about and the objectives.

The objective was stated in the last paragraph of the Introduction (page 5), last sentence: Response: "The objective of this study is therefore to explore how sociocultural factors may influence the scale up of the community-based distribution of injectable contraceptives in Nigeria."

3. The authors didn't also reported the overall rate of contraceptive use in Nigeria and the rate of injectable contraceptive.

Response: This has been added from NDHS 2018 (page 5, paragraph 1).

4. Line18-19: Of the three authors.....this sentence is unnecessary.

Response: The statement has been deleted as advised

Reviewer 2 (Emily Namey)

General comments

1. A strength of the research is that is it based on an existing program, providing the stakeholders who participated a discrete experience and set of observations to reflect upon, rather than speaking solely about hypothetical responses they or their communities may have to an intervention. This strength could be more clearly explicated and highlighted in the manuscript.

Response: This thought has been added to the Limitation and strengths section on pages 3 and 16 (paragraph1).

2. The primary weakness of the current draft of the manuscript is in the description of the methods and analyses, which are opaque and reference citations that do not show convincing appreciation for the empirical evidence or processes involved. I have provided specific questions and suggestions in the attached document.

Response: The Methods section has been updated largely using feedback from the reviewers. These updates are detailed in the specific comments below.

Specific comments

Page 4

3. Ln 22-24 “The sociocultural context may promote health or be a barrier to health (care) utilization
Response: Omitted word inserted as suggested (page 4, paragraph 2).

4. Ln 27-31 “Without due consideration of the sociocultural context, health innovation development, delivery and scale up is likely to be limited, if not dangerous!12”

The claim of potential danger seems undeveloped; unclear whether the citation supports this. Suggest explaining briefly (dangerous to whom? Why?), re-wording, and/or removing the exclamation point.

Perhaps move text from Page 5, lines 9-18, as these further explain the point/purpose

Response: This sentence has been re-worded as advised. The text from page 5 has also been moved to come immediately after the sentence as suggested. See page 4 (paragraph 2):” Without due consideration of the sociocultural context, health innovation development, delivery and scale up is likely to be limited, and this may lead to waste of resources including time.”

PAGE 5

2.1 Study design

5. Was anything published on the larger study of which this research was a part that you could reference?

Response: Yes, publications from the larger project have been cited, references 23 and 24 (section 2.1. page 5).

6. In either case, consider adding another sentence about what that entailed, which might help to justify the sample size and selection.

Response: A sentence has been added about the aim of the larger project – “...part of a larger research project^{23 24} which explored the interplay between barriers and facilitators of scale up of the community-based distribution of injectable contraceptives guided by the AIDED model³ in order to inform wider scale up of this innovation in Nigeria.” (Page 5) Justification for the sample size could be found in the ‘Participants and sample’ section on page 6.

7. I’m curious about the timing of the study relative to scale up. Seems to have happened after scale up ...were there issues/problems that led to the study? Or just the way the funding worked out? Lines 22-25 seem to suggest that this is a forward-looking approach to what may affect scale up, but the timing suggests perhaps a retrospective look at what has affected scale up.

Response: This study adopted both a retrospective and prospective approaches to assessing what affected or may affect scale up respectively. See timeline presented in Figure 1 as well as text on page 5 “...This study took place after scale up started, thus it adopted both a retrospective slant to assess factors that affected scale up and forward-looking approach to what may affect future scale up effort...”.

Participants and Sample

8. I’d like to see a clearer description in the narrative of which participants were interviewed and which in focus groups. The table provides detail which seems to indicate HCWs had IDIs, community members had FGDs, and senior stakeholders had KIIs. Also, the table seems to suggest one FGD each with different types of participants. If that is the case, it may be a very lean sample from which to draw conclusions and more will need to be said about why this was the selected sampling stratification (e.g., you were aiming to quickly assess the range of potential concerns, rather than understand the relative frequency/saliency of issues).

Response: The description of which participant group had which type of qualitative data collection method has been made clearer on page 7. Some of the suggestions here have been incorporated: “KIIS, IDIs and FGDs were conducted with senior stakeholders, healthcare workers and community members, respectively.” Earlier on page 6: “One FGD each was conducted with different types of participants who were involved and could give an account based on their experiences of the program with the aim of exploring qualitatively the views, concerns from this range of key role players, not aiming to quantify, compare or rank frequency of views. Each of the FGDs had eight participants.” (paragraph 2).

9. It seems many of the questions on the guides are asking participants to summarize the views of others, rather than offer their own view or experience. This might be used as a rationale for

conducting focus groups, which are typically helpful at getting at social norms.

Response: "The questions on the guides mainly asked participants to share their own experience and views as well as summarize the views of others...". (page 6, paragraph 2).

10. Also, in describing the setting, since religious values are discussed extensively later, it would be helpful to know if this is a predominantly Muslim community as readers may assume.

Response: Yes, Gombe State is a predominantly Muslim community. This has been added to the Study design and setting section: "The project took place in Gombe (North eastern Nigeria), one of 36 States in Nigeria and a predominantly Muslim community..." (page 5).

PAGE 6

11. As above, I'd like to see a more robust discussion of sampling. Justification of sample size could be better cited; there are several empirical studies that help to establish adequate sample size to reach saturation that would strengthen the authors' case for whether/why the selected sample was sufficient. Guest et al. (2006 and 2016) provide tables summarizing empirical literature on sample sizes and present empirical data of their own for interviews and focus group sample sizes.

Response: Thanks for these additional citations). These have been incorporated (references 31 – 33) to make the justification for the sample size more robust (page 6). The section on "Participants and sample" has been updated to include better justification of sample size (page 6).

12. It would be helpful to the reader to call out the average (or minimum) size of focus groups.

Response: Each of the FGDs had eight participants. This information has been added to the Participants and sample section (page 6).

13. It's possible (and common) to conduct an inductive thematic analysis without using grounded theory. Grounded theory is specifically indicated when the purpose of the research question is to develop a theory about a phenomenon. To my sense, the research question here is an applied, rather than theoretical, question: what are the specific issues or behaviors that could affect scale up? In this case, grounded theory is likely not warranted and my guess is that the constant-comparison method called for by grounded theory was not used. In any case, there are neither appropriate citations for grounded theory nor inductive thematic analysis. Check Glaser and Strauss, Charmaz, or Maman for grounded theory; check Guest et al. or Bernard for inductive thematic analysis.

Response: I agree that this is an inductive thematic analysis rather than grounded theory as theory development was not the purpose of our study. Ours was an inductive thematic analysis since it gives room to the views of participants, from where themes are identified and codes developed. Please see page 7.

2.2. Data collection

14. This paragraph isn't very clear in distinguishing what was asked of whom and in which format.

Were community members involved in FGDs? Line 33 refers to "interview guides used for discussions with community members" which I imagine are focus groups and therefore, according to the sentence prior, followed focus group guides. You could just say "all guides were translated..." to avoid confusion.

Response: "This has been clarified as suggested as "all guides were translated..." (section 2.2, page 7)

15. It would be helpful to clarify what was asked of all participants and what was asked specifically of one group or another. As it is, it seems all of the same topics were covered by all types of participants, which seems unlikely [supplemental file is helpful, but reader should have a grasp without reviewing those].

Response: Interviews with key stakeholders covered issues of program planning and design, community entry, and resolution of challenges with implementation. However, FGDs dealt with issues of program implementation, community members preferences with innovation delivery and packaging. Nonetheless, some issues covered in the interviews and discussions (FGDs) were similar, hence the authors believe it's better to summarize the contents together.

16. Ln 44-45 Was there any kind of transcription protocol? Were the translations and transcriptions

done by the data collectors or different people? Helpful to know a little more about the process so the reader can get a sense of data accuracy and consistency across transcripts.

Response: Yes, there was a transcription protocol. The process of transcription, translation and back translation has been described in this section (section 2.2, page 7): "...Transcription was done using a protocol which emphasizes verbatim transcription of recordings. Both transcription and translation were done by the same research associate who is vast in both Hausa and English languages. Back translation was subsequently done by another associate vast in the two languages; the back translations were afterwards compared with the original transcripts in order to ensure that the original meaning is retained.³⁶"

2.3 Data Analysis

17. I have briefly reviewed references 24 and 25 and neither seems to align with the description of the analysis process presented here. Specifically, article 24 relates to translation and reflexivity, and is very much a conceptual rather than processual piece. Article 25 outlines some of the tenets of grounded theory and walks through their use with CAQDAS, but it doesn't mention a "thematic framework approach". The description of pre-defined "thematic areas" sounds like they are not really thematic areas at all, but rather they reference the domains of inquiry, or what I would call "structural codes" – areas marked in the transcripts according to what was asked of the participant at that point.

Response: More appropriate references like Guest and Bernard have been added to replace references 24 and 25. The description of the analysis has been modified as "inductive thematic approach" (page 7, section 2.3)

18. This description of the analysis of emergent themes is somewhat weak. Who performed coding? At what level(s)? Was there any inter-coder reliability assessment?

Response: "Coding were done by the first and last authors independently at first before analysis started; these codes were then compared and harmonized." (page 7, section 2.3)

19. The statement that "Interviews were stopped at the point of saturation, where no new information was obtained.²⁴" is unclear. Was the sample set at the outset, based on empirical guidance on sample size relative to saturation, or was the aim all along to sample until saturation was reached. Was analysis really progressing alongside data collection, or were all data collected and then analysis began? It is not clear from either the data collection or analysis sections what the sequence of events was.

Response: Yes. Analysis progressed alongside data collection. A clearer and more detailed description of the process of data collection and analysis is provided on page 8, paragraph 1.

20. Also, related to saturation, was it that no new information was gained from anyone? Was there any comparative analysis of the different groups? Was the reaching of saturation predicated on the FGDs alone?

Response: " Although number of interviews and FGDs were set at the outset based on empirical guidance, however during analysis, it was confirmed that saturation,³⁰ where no new information was obtained, was actually reached before the cap was reached in both interviews and FGDs." (section 2.3, pages 7-8).

PAGE 7

21. Lns 6-9 "The authors reviewed the transcripts and compared themes, before these were applied to subsequent coding of all the transcripts." This part of the process is also a bit unclear. Is this meant to say that transcripts were reviewed prior to the development of a codebook? Was a codebook then developed? After how many interviews/FGDs? Or was all data collected (same Q as above) and then a codebook developed? It seems unusual to declare saturation reached (and to stop data collection) before coding is completed.

Response: " A codebook was developed after all the data were collected; this was then applied to the transcripts." (page 7, section 2.3)

RESULTS

22. I would like to see some summary of the demographic characteristics of the participants leading this section, particularly the FGD participants, so the reader has some way to interpret better from whom the subsequent findings/quotes come. For the KIIs and IDIs, it may suffice to reference broad general categories as in Table 1. For FGDs, gender and median age of participants would be useful
Response: The Results section is now started with sociodemographic characteristics of FGD and KII respondents (page 8). Table 1 has also been augmented.

23. Figure 1 is, I think, a summary of the three main overall findings. Each of the main areas to be discussed in the Results section should be identified in the narrative introduction to the section, so they align with the headings that follow and don't require the reader to process Figure 1 to understand what is coming.

Response: This suggestion has been incorporated (page 8)

24. 3.1 – here and throughout, where you say “participants” indicate which type of participants. Was this shared among all categories? The quotes for the first several findings seem drawn only from IDIs/KIIs – was this because only they were asked or because these themes did not come up in FGDs?

Response: As much as possible, "participants" have been qualified whether interview or FGD. KII and IDI participants tended to openly highlight religious and cultural beliefs as an issue while FGD participants tended to bring up issues of cultural practices / beliefs a bit indirectly (e.g. men control what wives can do but it's because they are unaware?)

25. Also, throughout, do you have the possibility of indicating the frequency of statements in support of the finding you're presenting? In the cases where you indicate “one person said” that's helpful to know; where it's not specified, it's hard to know if this was a frequently repeated theme or a one-off idea

Response: We are of the humble opinion that presenting frequencies against statements defeats the purpose of a qualitative study and makes the data look more like a quantitative study without probability sampling.

26. Ln 52-54 copy edit “Emerging from this world view, participants described a subtle resistance” in this phrasing, the participants themselves were emerging from this world view, when I think you mean the resistance was related to the world view.

Response: The phrasing has been modified to "Related to this world view..." (page 9, paragraph 1).

27. For section 3.3. was this asked only of FGD participants? Could you do a comparative analysis of responses? Were current users generally in favor and men mixed? An issue is that the FGD is the unit of analysis, so you only have 10 and it's hard to make determinations about whether a group is for/against something.

Response: Yes this question was explored only with FGD participants, and the CBD approach was supported by women who currently use contraceptives and married men. Although there was a dissenting opinion among the married men which was captured.

We really do think we should desist from reporting quantitatively on which group is for or against – this was not the aim of the study and study not set up this way. The focus of the study was to explore which perceptions were raised in the FGDs – with the aim of reporting which were potential facilitating and which potentially inhibitors to CBDIC scale -up, and later discussing in which ways these issues could influence scale up.

28. In some sections, like 3.5, it seems only one category of participant is represented. Were only traditional/religious leaders asked these questions about community support? If so, state that clearly

Response: Yes, the issue of community support for scale up was solely explored with community and religious leaders. This has been stated clearly: “The issue of community support for scale up was solely explored with community and religious leaders.” (page 12).

PAGE 14

29. 4.1 Limitations – This section should be revisited. The sample for this study was extremely small and, as yet, unjustified in its size and scope. "... findings are not statistically generalizable but the theories and ideas can be applied to other contexts." It's not clear that these theories and ideas can be applied in other contexts with any confidence, based on the study design and data alone...perhaps more accurately, the findings are in line with what others have found in similar (religious/LMIC) settings, which adds to the collective body of knowledge on socio-cultural issues that affect contraceptive uptake.

Response: This section has been re-written: "... a strength of this study is that it provided stakeholders who participated a distinct experience and some observations to reflect upon since the study was based on a current program, rather than speaking merely about theoretical responses they or their community members may have to a health intervention." (page 16, paragraph 1).

4.2 Recommendations

30. "Further studies are needed on the barriers and facilitators to community-based distribution of health innovations.⁴¹ Although the pilot project¹⁶ provided some evidence that community based distribution is feasible in Nigeria, more understanding about what may impede or enhance this approach to health commodity distribution is needed." This statement is fairly generic. It seems you cited a number of studies on barriers and facilitators, and listed several things that stakeholders need to consider. So what, specifically, are questions that need further research?

Response: The generic part of the Recommendation has been deleted and the remaining part combined with the Conclusion segment (page 16)

PAGE 15

4.3 Conclusions

31. There is a lot of repetition between the Discussion and Conclusion sections. Suggest focusing the Discussion section on situating the findings within the broader literature and the Conclusions on what the findings mean in terms of next steps for research and/or practice.

Response: This section has been re-structured as advised (page 16). The repetitions have been deleted from the Conclusion section. Implications of the findings have also been incorporated in this section

Reviewer 3 (Margaret Gichane)

General comment

The study is described as using a grounded theory approach, however the analysis described does not align with grounded theory.

Response: This has been modified as inductive thematic analysis (page 6)

Abstract

1. Conclusion needs to more align with the findings. Include a sentence summarizing findings

Response: The Conclusion has been modified to focus on what the findings mean for practice and research, as suggested by Reviewer 2.

Introduction

2. The introduction does not include any description on the current evidence on community based distribution of injectable contraceptives in other countries in Africa. Consider including...

Response: The suggested evidence from Africa has been incorporated (references 16 -19).

3. Include information on who conducted the interviews

Response: This has been incorporated – " The interviews and FGDs were conducted by the first

author assisted by three trained research assistants.” (page 6, paragraph 2)

4. Page 5, Line 32: Describe the parent project in more detail. Additionally, include which stage in the research study the interviews were collected. If it was after study activities had commenced include in the limitations how that may have affected findings.

Response: This has been done “This cross-sectional study conducted from September to November 2016, part of a larger research project^{19 20} which explored the interplay between barriers and facilitators of scale up of the community-based distribution of injectable contraceptives guided by the AIDED model³ in order to inform wider scale up of this innovation in Nigeria.” (page 5, section 2.1) A timeline to show when data collection was done has also been added (see Figure 1)

5. Page 6, line 12: change “a point where no new data is being gotten” to “no new information is emerging.”

Response: This has been modified (page 6, paragraph 2).

6. Page 6, line 30: Change, “Interview guide and focus group guide were used “ to “Guides were used...”

Response: Modified (page 7)

7. Page 6, line 52: The data analysis is described as a “thematic framework approach” but the study was classified as using a “grounded theory approach.” Grounded theory follows a specific set of steps when coding that were not part of this study. See research from qualitative methodologists such Charmaz, Strauss and Glaser. It is more appropriate to use just name this as a qualitative study that used a “thematic framework approach” for analysis

Response: This has been modified as “inductive thematic approach” (page 6)

8. Table 1. Though it is correct to say the “n” for FGDs refers to number of groups not individual participants it makes it difficult to compare the table to the text (n=32 interviews vs. 102 participants). Consider including a column where you include ns for participants. For IDIs the n will be the same in both columns but for FGDs one column will have the number of groups and the other will have the total number of participants in groups.

Response: Table 1 has been modified with an additional column, as advised

9. Page 13, Line 15: Indicate who is passively resisting. Is it women? Husbands?

Response: This has been incorporated- “both women and their spouses” (page 14, paragraph 2)

10. Page 14, line 27: It is unclear what finding this paragraph is related to. Which solutions were proposed?

Response: This refers to solutions proffered by participants and those inferred from the Discussion

11. Page 14, line 42: An additional limitation is that you recruited some participants who were involved in a community based distribution intervention therefore they may provide more socially desirable answers.

Response: This limitation has been incorporated (page 2 and page15).

12. Page 14, line 54: These recommendations seem very general and not related to the findings. The results point to the importance of considering patriarchal views and religiosity when designing CBD programs. It seems like interventions with men should be recommended

Response: Recommendation modified and merged with Conclusion- “Interventions to improve male participation are recommended with the aim to improve the scale up process.”

VERSION 2 – REVIEW

REVIEWER	Margaret Gichane RTI International
REVIEW RETURNED	26-Feb-2020
GENERAL COMMENTS	Just a note, for future resubmissions, please include a cover letter where you respond to each reviewer comment individually and note how you addressed the comment in the manuscript. This is

	standard practice in reviews and makes it easier for reviewers to assess whether suggested changes were adequately addressed. The authors adequately addressed my concerns and I approve the study for publication. One minor edit, grounded theory should be removed from the “design” section of the abstract.
--	--

VERSION 2 – AUTHOR RESPONSE

As advised by reviewer 3, "grounded theory" has been deleted from the Design section of the Abstract (page 2) and replaced with "inductive thematic analysis".